# One size does not fit all: Nurturing identity needs and job satisfaction through employee benefits across gender and age

**Alessia Valmori**[1]*, **Eleonora Reverberi**[2], **Claudia Manzi**[1]

**1** Catholic University of the Sacred Heart, Milan, Italy, **2** Jointly Il Welfare Condiviso s.r.l., Milan, Italy

\* alessia.valmori1@unicatt.it

## Abstract

The impact of employee benefits on workers' identity needs was examined through an integrated framework combining social exchange theory with identity-based mechanisms. Two large-scale studies (Study 1: $N = 13368$, $N_{Firms} = 5$; Study 2: $N = 6337$, $N_{Firms} = 3$) investigated how the quantity (number of benefits used) and quality (satisfaction with benefits) of organizational benefits affect workers' identity needs across four welfare categories: Work-life integration, health/safety, financial, and socio/cultural. Results consistently showed that benefits quality, rather than quantity, are more strongly associated both identity needs and job satisfaction. The impact of benefits varied according to benefit categories and demographic characteristics, suggesting the importance of considering individual differences in designing effective benefits programs.

## Introduction

Employee benefits refer to the efforts made by employers to improve the working and living conditions over and above the wage [1]. In the last decades, both public and private sector organisations have increasingly provided welfare facilities such as housing, medical benefits, and education [2]. Notably, these benefits convey the fact that they are designed not to create direct economic value for the enterprise, but to support employees' needs [3]. This raises the question: why should firms invest in employee benefits?

If we analyse employee benefits from an organizational perspective, a large-scale study conducted by Liang and colleagues [4] shows that there is a positive association between employee-friendly practices and firm performance. Moreover, when a company invest in employees' welfare it is likely to be perceived as more socially responsible by a market that is increasingly attentive to these aspects [5].

If we analyse employee benefits from the worker's perspective, it has been argued that employee benefits increase workers' well-being and job satisfaction

**Data availability statement:** We have included all datasets on OSF (https://osf.io/cbxwj/?view_only=6f2f7810809f43af9e0b6e76422b12f7).

**Funding:** This research was supported by grant PNRR PE00000015 of the European Union – NextGenerationEU, entitled 'Age-it, theme 8: Consequences and challenges of ageing'.

**Competing interests:** The authors have declared that no competing interests exist.

[6–8]. Society for Human Resource Management's report [9] showed that American employees evaluated organizational benefits as one of the top three incentives that increase their job satisfaction and intention to stay in the company.

Moreover, as the workforce population has increased in terms of diversity (gender, age, disability, etc.) [10], employee benefits programs can effectively address the needs of employees in their different socio-demographic profiles [11]. The impact of employee benefits on workers' psychological outcomes has been previously empirically tested [12–14] and it has been studied mainly through the theoretical framework of the social exchange theory [15]. Fewer researchers studied the positive effect that organizational benefits may have on workers' identity processes.

The studies we present here aim to address this latter aspect and to provide a first empirical test of the impact of the quantity (number of benefits used by the employer) and the quality (satisfaction with benefits used) of employee welfare on workers' identity needs as a function of their age and gender, using two large samples of employees. Study 1 tested the benefits' impact on workers' identity needs, considering age and gender moderation effects. Study 2 replicated Study 1 findings testing a mediation model in which the associations between job satisfaction and quantity and quality of employee benefits were mediated by identity needs. As a further aspect of novelty of our studies, we tested out model comparing four different categories of benefits: work-life integration, health/safety, financial, and socio/cultural.

### Employee benefits, workers' identity and job satisfaction

The relationship between employee benefits and organizational outcomes has traditionally been examined through the lens of social exchange theory [15–18]. This theoretical framework suggests that when one party provides a benefit, the recipient feels obligated to reciprocate, implying that employees develop a sense of obligation toward organizations that invest in their well-being through benefits programs [19].

However, mechanisms beyond simple social exchange may also explain how employee benefits translate into positive work outcomes. More recently, an identity-based perspective has emerged, suggesting that the employee-employer relationship is better understood through the lens of workers' identity processes rather than reciprocity or exchange [20]. For example, several studies have consistently demonstrated a positive relationship between Perceived Organizational Support (POS) and organizational identification [21]. From this identity perspective, scholars proposed and tested that POS fulfils employees' socio-emotional needs-such as esteem, approval, affiliation, meaning, and purpose-thereby promoting organizational identification [22,23]. Since individuals are motivated to maintain or enhance personal needs linked to social identification [24,25], they are more likely to identify with groups that fulfil these needs [24] and to develop strong involvement with such groups. In line with this reasoning, several scholars have argued that by meeting socio-emotional needs, POS increases the likelihood of employees identifying with their organization [26,27]. Building on findings from the POS literature, we theorize that organizational benefits influence work outcomes by shaping employees' identity-related processes [28,29].

We focus on job satisfaction as the primary outcome because it is linked to better job performance at individual [30] and organizational levels [31,32], and prior work shows organizational identification relates positively to this outcome [33].

## Unpacking different areas and dimensions of employee benefits

Under the broad term of employee benefits, we can find numbers of different services that have been clustered differently. In line with Fulmer & Li and Dulebohn et al. [34,35], we considered three clusters of employee rewards: work-life integration, health/safety, and financial benefits. Moreover, we also added benefits relative to another dimension that is crucial to sustaining individuals' identity needs, namely socio/cultural benefits [36].

Work-life integration benefits are aimed at supporting employees in balancing work and family responsibilities. In a systematic review of qualitative studies, Campbell and colleagues [37] show that employee benefits schemes, such as on-site childcare, paid leave, and flexible work hours, play a crucial role in enhancing the well-being and satisfaction of employees, particularly of parents with young children. More recently these findings have been replicated by Masterson and colleagues' [38] systematic review which shows that these benefits not only contribute to staff recruitment and retention, but also to reduced absenteeism, increased job satisfaction, and enhanced company loyalty.

Health services are a crucial component of employee benefits, aiming to improve health and wellness and prevent ill health across physical, social and psychological dimensions. Health/safety benefits, as part of welfare measures, include safety training and awareness programs, workplace safety initiatives, social security provisions, health insurance, and facilitated access to medical facilities [3]. These measures not only may contribute to employee retention but also enhance motivation and overall efficiency by promoting a safe and healthy work environment [39]. When health and safety organizational benefits were considered, Parks & Steelman's metanalysis [40] reported the positive effects of these benefits in reducing absenteeism and fostering job satisfaction.

Financial services in employee benefits encompass a variety of offerings aimed at enhancing employees' financial well-being. These services include retirement funds, financial coaching, payroll advances, short-term instalment loans, credit counselling, debt management, online financial management tools, etc. [41,42]. There is little study that has analysed the effect of this type of benefits disentangling them from compensation. One exception is a study on retirement funds which indicates that these benefits can enhance organizational commitment through dual mechanisms: by fostering emotional attachment to the company (affective commitment) and by creating practical incentives that encourage retention (continuance commitment) [43].

Finally, socio/cultural benefits aim at the intellectual and social recreation of employees through activities such as sports, cultural events, libraries, and volunteering activities, fostering a sense of community and personal growth both within and beyond the workplace [6]. These benefits also include physical spaces dedicated to the social interaction between co-workers such as relaxation areas and canteens. Very little is known about the positive effect of such kind of initiative on workers' well-being. An exception is a study by De Gilder and colleagues [44] which shows how companies' organization of volunteering activities for employees enhances employees' positive attitudes toward the job and the company.

Importantly, in all these areas, employee benefits may be beneficial for employees not only based on the number of services used but also on the quality of each benefit. Previous studies suggest that the number of services offered (quantitative indicators) should be analysed together with the users' satisfaction with those services (qualitative indicators) [45,46]. On one hand, just the availability of organizational benefits creates a virtuous cycle in employees. Indeed, Grover & Crooker [47] found that the mere availability of work-life benefits proposed by the firms increased employees' attachment, independently of whether the employees utilised these services. In some way, having these offers is a symbol that the company cares about the employees, so it could be sufficient for creating positive work outcomes. On the other hand, previous literature [18,48] demonstrated that the quality of the offer (measured as monetary investment organizations made in the benefits and workers satisfaction with the benefits used) is predictive of higher satisfaction compared to mere utilisation.

In the current project, we will focus on quantitative and qualitative indicators of welfare based on workers' report of these aspects. In particular quantitative indicators of welfare are analysed in terms of number of benefits of the different categories used, while qualitative indicators are measured as workers' satisfaction with the benefits used. We test whether these two aspects may influence employees' identity processes and in particular, we aim to extend the 'quality> quantity' claim by testing whether the advantage of benefit quality over mere utilisation holds across different welfare categories (work–life integration, health/safety, financial, socio/cultural) and across key sociodemographic profiles (age and gender). Breaking down the analyses by type of benefit and socio-demographic classes is an important innovative element that provides valuable insights to help organisations target scarce resources toward fewer, higher-quality benefits that reliably improve workers' outcomes for specific groups, avoiding costly, low-impact expansions in benefit quantity that may not translate into meaningful employee outcomes.

## Gender and age tailoring of employee benefits

As organisations employ an increasingly diverse workforce [49], it becomes crucial to evaluate how they support the welfare of all employees. Previous literature on the compensation-activation theory [50] already focused on a person-centred approach when compensation is considered, introducing the possibility of customizing pay practices to foster individual activation to their work. Indeed, the customization of organizational benefits through methods such as focus groups and employee surveys demonstrates organizational responsiveness to individual needs, which enhances employees' sense of being valued by their organization [16].

Concerning gender, previous studies indicate that women tend to appreciate more the benefit schemes offered by the firms compared to men [1] and that organisational support tends to be more effective for women [49]. In analysing the intersection between gender and different areas of welfare, based on previous findings, we can anticipate that women should benefit more from welfare initiatives related to health and safety needs [51] and socio-cultural activities [52], whereas men should benefit more from welfare initiative related to financial support [53]. As far as work-life balance, there are contradictory results. On one hand, women face additional challenges in this area as they are traditionally disproportionately charged with the care load [54], and thus we should expect a stronger impact of welfare initiatives in the area of work-life balance on women. On the other hand, some evidence shows that support for work-life balance has an equally beneficial effect for both men and women [55–57].

Concerning age, existing research examined age differences [11] reporting that healthcare benefits were more important for Boomers (older generation) while financial and work-family integration benefits were more important for Gen X and Gen Y (younger generations). Indeed, older workers may derive less benefit from work-life integration services than younger employees, as these measures are often more valuable to employees with younger children [58,59]. Additionally, social and cultural activities can be beneficial for older employees in terms of positive ageing [60,61].

To our knowledge, no study considers the intersection of gender and age. For this reason, in analysing the impact of employee benefits on workers' identity needs and job satisfaction we took into consideration both gender and age differences. Considering older women employees has become a crucial point of attention for the organisation. This group faces unique challenges arising from the intersection of both age and gender, including perceived competence issues, stigma, stereotyping, and limited promotional opportunities, all of which can impact their decision to exit the workforce [62,63]. Age and gender stereotypes can negatively impact their sense of authenticity, organisational identification, and self-assessed performance [64]. Therefore, in our studies, we also take into consideration the intersection between age and gender to unveil strategies to support women over 50.

## The current research

The current research project aims to extend previous literature that demonstrates the positive impact of employee benefits on job satisfaction, integrating an identity-based perspective.

Specifically, we test whether the quality of the offer is more explanative of the positive effect on worker's identity needs fulfilment compared to the mere utilisation of the offer across different types of benefits (i.e., work-life integration, health/safety, financial and socio/cultural benefits) and different socio-demographics of workers (i.e., gender and age). Moreover, we tested a mediational model in which identity needs mediates the relationship between different types of employee benefits and job satisfaction.

In Study 1 we focus on the following objectives:

O1: To examine the direct impact of quantitative and qualitative indicators across 4 categories of employee benefits, controlling for age and gender.

O2: To test for a possible moderating effect of gender and age.

O3: To test for the possible combined moderating effect of both age and gender.

In Study 2 we replicated the objectives of Study 1, focusing on specific workers' outcomes, namely job satisfaction. We formulated a fourth objective:

O4: To test a mediation model in which employees' identity needs mediates the effect of employee benefits on job satisfaction.

## Study 1

### Method

#### Participants

The data for this study were collected from participants working at 19 companies that have partnered with the company Jointly for the provision of welfare services. Jointly is a Certified B Corporation (B Corp®) specialized in welfare and corporate wellbeing, committed to ensuring a positive impact on the wellbeing of individuals, companies, and local communities.

The questionnaire was administered from April 2024 to June 2024 by the research team in collaboration with Jointly to the entire workers population of the 19 companies involved. The companies operated across various sectors, including transportation, insurance, food, cooperatives, credit, e-commerce, manufacturing, film production, business services, and telecommunications. Thirteen companies had between 100 and 5000 employees, five companies had between 5000 and 50000 employees, and only one firm had over 50000 employees. Participants initially provided information about their job roles and then shared their opinions on welfare services usage and satisfaction.

Subsequently, they completed a measure of identity needs satisfaction. Finally, respondents were asked to provide sociodemographic information such as gender and age. The questionnaire could be completed via PC, smartphone, or tablet. This study was granted ethics approval by the local committee at the University of Padova and all participants involved provided a written informed consent. To reduce potential method biases resulting from self-report measurements, such as giving socially desirable responses to reduce the fear of being judged, following Podsakoff et al. [65], the interviewees remained anonymous and they were assured that there were no good or bad answers, and were asked to be as sincere and honest as possible.

Data from 13368 employees across 5 firms were considered.

#### Analytical sample

For our analysis, most participants did not use at least one service from each benefit category, we also included them in the analysis, allowing the sample size (N) for each model to vary based on the number of participants who used the specific service considered in the model.

Table 1 shows the distribution of the participants across gender and age groups.

**Table 1. Frequency Distributions of Participants (N = 13368) in Study 1.**

|  | Women | Men | Total |
|---|---|---|---|
| Under 50 years | 2184 (16%) | 4643 (35%) | 6827 (51%) |
| Over 50 years | 1695 (13%) | 4846 (36%) | 6541 (49%) |
| Total | 3879 (29%) | 9489 (71%) | 13368 |

## Procedure

**Variables. Quantitative indicators of welfare.** Based on previous work on employee benefits [16,66,67], we assessed the quantitative use of the benefits. The measure included 47 items assessing whether the employees used the employee benefits across the 4 areas under investigation. Fifteen items were related to work-life balance (e.g., school orientation for employee's kids, babysitting, the possibility of bringing children to the office), 11 items to health/safety benefits (e.g., flu vaccination in the organisation, psychological support, gym), 15 items to financial services (e.g., retirement fund service, credit facilities/agreements with insurance products) and 6 items to socio/cultural opportunities (e.g., relax/comfort rooms, corporate volunteering) and participants indicated whether they had used each benefit (1) or not (0). We calculated a sum score across the 4 areas of welfare investigated to achieve a quantitative indicator of welfare on work-life balance, financial, socio/cultural and health/safety categories.

**Qualitative indicators of welfare.** Following previous work on employee benefits [18,68], we assessed the employees' satisfaction with the benefits they used. The measure included 47 items assessing employee satisfaction with the service they utilised, rated on a scale from (1) *Not at all satisfied* to (5) *Completely satisfied* across the 4 areas under investigation. A single average score was then calculated for each of the 4 categories of benefits.

**Identity needs.** Six items were administered aimed at assessing employees' fulfilment of their identity motives within the firm adapted from Manzi et al. [23]. Each item captured a different dimension of the motivated identity construction theory, namely self-esteem, continuity, meaning, distinctiveness, self-efficacy and belonging [69]. An example of an item is: "Being part of X gives me a sense of belonging". The items showed higher reliability, Cronbach's α = .89, thus for the final analysis we computed an average score of the six items.

## Data analysis strategy and methods

First, we preliminarily explored whether there were differences related to gender and age in the quantitative and qualitative indicators of the different categories of benefits and with the identity needs. These results are fully reported in the Supporting Information.

Second, we tested the effect of quantitative and qualitative indicators of employee benefits on employees' identity needs within the firm. In doing so we also analysed for a possible moderation effect of gender and age group. Given the nature of our data, we decided to test the model at the individual level and control for any effect of the company by performing multilevel analyses. To investigate this, we employed three different linear mixed models for each of the four categories of benefits utilising the GAMLj module in jamovi version 2.3.28.0 [70]. In Model 1 we included the quantitative and qualitative indicators, controlling for gender (categorised as men and women) and age (differentiating under and over 50) as fixed factors. Additionally, we accounted for the company ID as a random factor. In Model 2, we tested the interaction of the quantitative and qualitative indicators with gender and age. In Model 3 we also explored the three-way interaction assessing a combined moderation effect of age and gender. Tables reporting the ANOVA omnibus effects are reported as Supporting Information (S10 – S13 in S1 File)

For every analysis, although data from the models were not normally distributed (*Kolmogorov-Smirnov* normality of residuals test *p* < .001), we still decided to employ parametric statistical models. In line with Bilon [71], we considered that with large samples normal distribution tests may generate false positive responses.

## Results

**O1: Examining the direct impact of quantitative and qualitative indicators of 4 categories of employee benefits after controlling for age and gender (Model 1).** The quality of the work-life integration (*b* = .23, SE = .02, *t* = 9.10, *p* < .001), health/safety (*b* = .17, SE = .03, *t* = 5.88, *p* < .001), financial (*b* = .25, SE = .01, *t* = 24.77, *p* < .001), and socio-cultural (*b* = .16, SE = .01, *t* = 15.94, *p* < .001) benefits used was associated with higher identity needs fulfilment. Only the quantity of financial (*b* = .05, SE = .01, *t* = 3.53, *p* < .001) and socio-cultural (*b* = .04, SE = .02, *t* = 2.31, *p* = .02) benefits was associated with higher identity needs fulfilment.

**O2: Testing for a possible moderating effect of gender and age (Model 2).** The moderating effect of age or gender emerged only for health/safety and financial benefits.

Concerning the health/safety benefits, the two-way interaction between age and the quantitative indicator of the health/safety benefits emerged. As shown in Fig 1 A, the simple effect analysis revealed that, although the tendencies are not significant, the association of the quantity of health/safety benefits with the identity needs showed different trends for those under 50 (*b* = −.04 3, SE = .05, *t* = −.84, *p* = .40) and over 50 employees (*b* = .14, SE = .09, *t* = 1.62, *p* = .11).

Regarding the financial benefits, the two-way interaction between gender and the qualitative indicator of financial benefits emerged from the analysis. As shown in Fig 1 B, the quality of the financial benefits was more strongly associated with men's identity needs (*b* = .27, SE = .01, *t* = 21.9, *p* < .001) compared to women (*b* = .22, SE = .02, *t* = 12.3, *p* < .001).

**O3: Testing for the possible combined moderating effect of both age and gender (Model 3).** The moderating effect of age and gender emerged only for the socio/cultural benefits. When the qualitative indicator of the social and cultural benefits used was considered, the three-way interaction with gender and age emerged. The simple effects analysis revealed that, although for men (*b* = .17, SE = .02, *t* = 10.15, *p* < .001) and women (*b* = .16, SE = .02, *t* = 6.62, *p* < .001) under 50 the association of the quality of the social and cultural benefits with identity needs is similar (Fig 1 C 1), for employees over 50 the quality of the social and cultural benefits was more strongly associated with identity needs for women (*b* = .24, SE = .03, *t* = 8.20, *p* < .001) compared to men (*b* = .13, SE = .02, *t* = 7.45, *p* < .001) (Fig 1 C 2).

## Discussion

Study 1 aimed to test the effect of different types of employee benefits on employees' identity needs. Specifically, we examined whether the quantity of benefits (measured by benefits use) or the quality of benefits (measured by satisfaction with benefits) was more strongly associated with identity needs fulfilment. We also tested the moderating effects of gender and age.

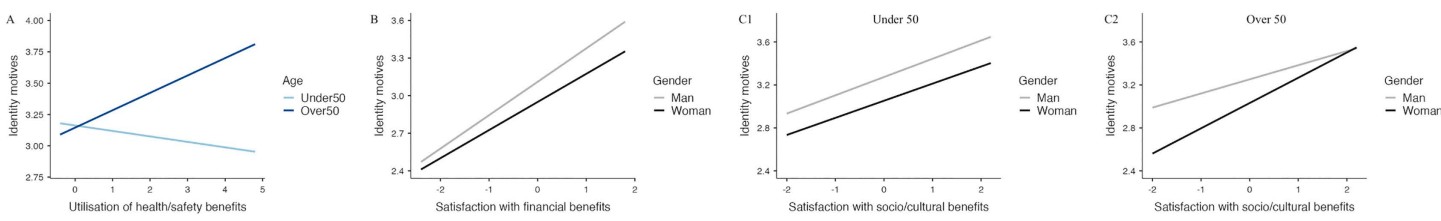

**Fig 1. The Moderating Effect in Study 1.**

Our findings suggest that qualitative indicators have a greater impact than quantitative indicators in predicting employees' identity needs fulfilment. We also found some significant moderation effects of gender and age. Specifically, the quality of health and safety services seems to be more strongly associated on employees over 50 compared to those under 50 whereas the quality of the financial services was more strongly associated with identity needs for men compared to women. Importantly, the quality of socio-cultural benefits seems to be particularly important to women over-50 compared to other socio-demographic groups.

Taken together, these results suggest that firms should consider not only the quantity of services offered but also the quality to effectively impact on employees' identity needs, as it is revealed to be associated with employees' identity needs.

Identity needs fulfilment is an important aspect to take into consideration, as previous literature has shown that these needs significantly enhance employees' job satisfaction [72,73]. In Study 2, we aimed to replicate the findings from Study 1 by also testing the mediating effect of identity needs on the relationship between quantitative and qualitative indicators of welfare and employees' job satisfaction.

## Study 2

### Method

**Participants.**  This study gathered data from employees across three different companies, all of which had partnered with Jointly. Data were collected from April 2024 to June 2024. The same procedure and a similar survey as for Study 1 was employed in Study 2. The only difference was the inclusion of a measure of job satisfaction.

Data from 6337 employees across 3 different firms were considered.

**Analytical sample.**  Although most participants did not use at least one service from each benefit category, we included all these participants in the analysis. Consequently, the sample size (N) for each model varies based on the number of participants who used the specific service taken into account in the model.

Table 2 shows the distribution of the participants across gender and age groups.

### Procedure

**Variables.  Quantitative indicators of welfare.** The measure included 90 items assessing whether the employees used the employee benefits in the 4 areas under investigation. Thirty-nine items were related to work-life balance (e.g., hybrid working, babysitting), 19 items to health/safety benefits (e.g., health insurance, psychological support), 26 items to financial services (e.g., public transportation discounts, retirement funds), and 6 items to socio/cultural opportunities (e.g., relaxation/comfort rooms, corporate volunteering). Participants indicated whether they had used each of the benefits (1) or not (0). We calculated a sum score across the 4 areas of welfare investigated to achieve a quantitative indicator of welfare on work-life balance, financial, socio/cultural and health/safety categories.

Table 2.  Frequency Distributions of Participants (N = 6337) in Study 2.

|  | Women | Men | Total |
|---|---|---|---|
| Under 50 years | 2265 (36%) | 1721 (27%) | 3986 (63%) |
| Over 50 years | 1123 (18%) | 1228 (19%) | 2351 (37%) |
| Total | 3388 (53%) | 2949 (46%) | 6337 |

**Qualitative indicators of welfare.** The measure included 90 items assessing how much employees were satisfied with the service they utilised on a scale from (1) *Not at all satisfied* to (5) *Completely satisfied* in the 4 areas under investigation. A single average score was then calculated for each of the 4 categories of benefits.

**Identity needs.** As for Study 1, an average score based on the six items adapted from Manzi et al. [23] assessing employees' identity motives fulfilment within the firm was computed (Cronbach's α = .91).

**Job satisfaction.** One item was considered to assess participants' satisfaction with their job: "All in all, how satisfied do you feel with your work life in general?". Participants replied on a scale from (1) *Completely dissatisfied* to (10) *Completely satisfied*.

## Data analysis strategy and methods

In the first phase, we performed the same analysis we conducted in Study 1 to test the replicability of the results in a different sample. These analyses are fully reported in the Supporting Information.

Second, we tested our quantitative and qualitative model of employee benefits as we did for Study 1. Tables reporting the ANOVA omnibus effects are reported as Supporting Information (S24 – S27 in S1 File)

Finally, we tested the mediational effect of identity needs in the relationship between quantitative and qualitative indicators of the different categories of benefits and job satisfaction. We performed four path analyses for each category of benefits (i.e., work-life integration, financial, socio/cultural and health/safety benefits) utilising the PATHj module in jamovi version 2.3.28.0 [74]. Specifically, we included the quantitative and qualitative indicators of the specific category of employee benefits as exogenous variables associated with identity needs and in turn with job satisfaction. We also added age and gender as controlling variables of both the first and the second path. For each model, we included the firm as a group variable and constrained the model to compute equal regression coefficients across the groups. Since no significant differences emerged between firms, we reported the results based on the combined data. Fig 2 illustrates the tested models. We reported the indirect effects of both the quantitative and qualitative indicators of each benefit category on job satisfaction through identity needs.

## Results

**O1: Examining the direct impact of quantitative and qualitative indicators of 4 categories of employee benefits after controlling for age and gender (Model 1).** Across the four categories of benefits, the quality of the work-life integration ($b$ = .23, SE = .02, $t$ = 12.36, $p$ < .001), health/safety ($b$ = .19, SE = .03, $t$ = 7.53, $p$ < .001), financial ($b$ = .35, SE = .03, $t$ = 13.84 $p$ < .001), and socio-cultural ($b$ = .12, SE = .02, $t$ = 5.08, $p$ < .001) benefits was associated with higher

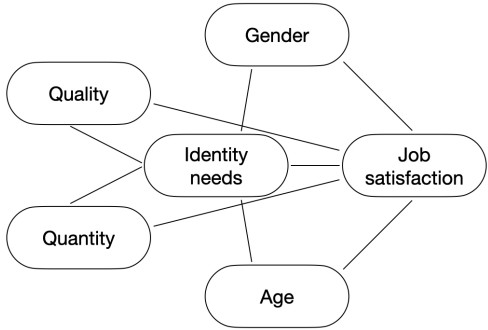

**Fig 2. Representation of the Path Analysis Models Performed in Study 2.**

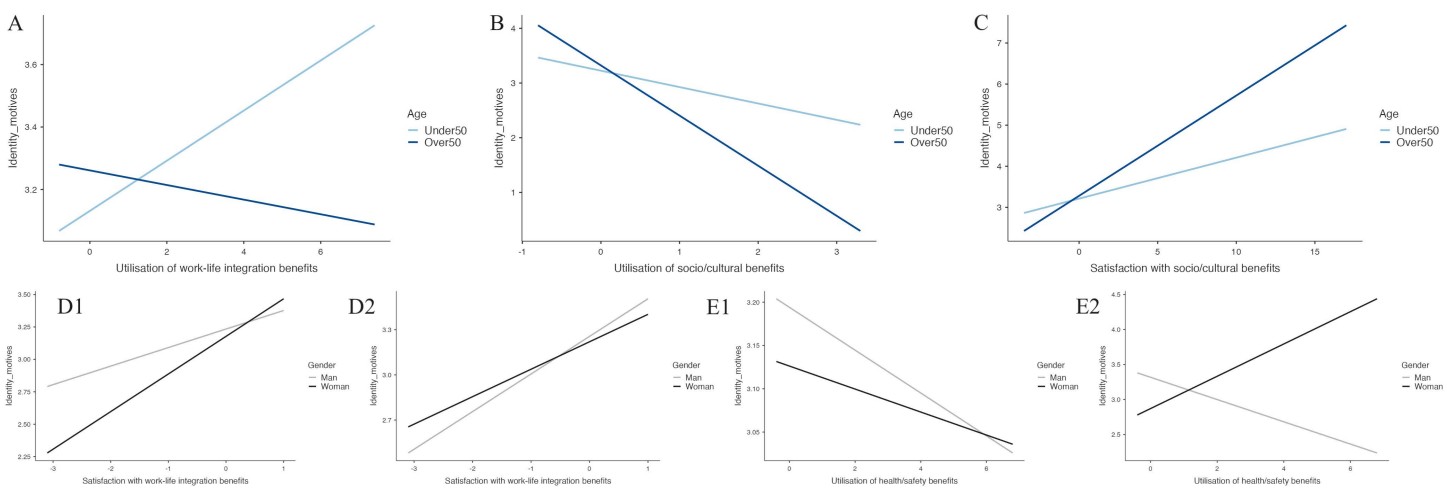 *(PLOS One logo in header area)*

identity needs fulfilment, compared to the mere utilisation of them. The quantity of the work-life integration benefits was also positively associated with employees' identity needs ($b = .04$, SE $= .02$, $t = 2.34$, $p = .02$). Notably and contrary to what emerged in Study 1, when socio/cultural benefits were considered, the utilisation of them was negatively associated with employees' identity needs fulfilment ($b = -.04$, SE $= .09$, $t = -4.03$, $p < .001$).

**O2: Testing for a possible moderating effect of gender and age (Model 2).** The moderating effect of age or gender emerged only for work-life integration and socio/cultural benefits.

Concerning the work-life integration benefits, the two-way interaction between age and utilisation of work-life integration benefits emerged. As shown in Fig 3 A, the quantity of work-life integration benefits was positively associated with identity needs only for those under 50 ($b = .08$, SE $= .02$, $t = 3.55$, $p < .001$) compared to those over 50 ($b = -.02$, SE $= .03$, $t = -.91$, $p = .37$).

Regarding socio/cultural benefits, the interaction between age and quantitative indicators of the socio/cultural benefits emerged (Fig 3 B). The simple slope analysis revealed that the quantity of the social and cultural benefits was more negatively associated with the identity needs of over 50 employees ($b = -.91$, SE $= .26$, $t = -3.50$, $p < .001$) compared to under 50 employees ($b = -.30$, SE $= .10$, $t = -2.95$, $p = .003$).

Also, the two-way interaction between age and the qualitative indicator of the socio/cultural benefits emerged. As shown in Fig 3 C, the quality of the social and cultural benefits used was more strongly associated with identity needs on over 50 employees ($b = .24$, SE $= .06$, $t = 3.75$, $p < .001$) compared to those under 50 ($b = .10$, SE $= .03$, $t = 3.96$, $p < .001$).

**O3: Testing for the possible combined moderating effect of both age and gender (Model 3).** The moderating effect of age and gender emerged only for work-life integration and health/safety benefits.

When the qualitative indicator of the work-life integration benefits used was considered, the three-way interaction with gender and age emerged. The simple effects analysis revealed that, although for over 50 employees the qualitative indicator of work-life integration benefits was associated with identity needs slightly more for men ($b = .25$, SE $= .05$, $t = 5.49$, $p < .001$) compared to women ($b = .18$, SE $= .04$, $t = 4.18$, $p < .001$) (Fig 3 D 1), when under 50 are considered, the quality of the work-life integration benefits was more strongly associated with identity needs for women ($b = .29$, SE $= .03$, $t = 9.91$, $p < .001$) compared to men ($b = .14$, SE $= .04$, $t = 3.86$, $p < .001$) (Fig 3 D 2).

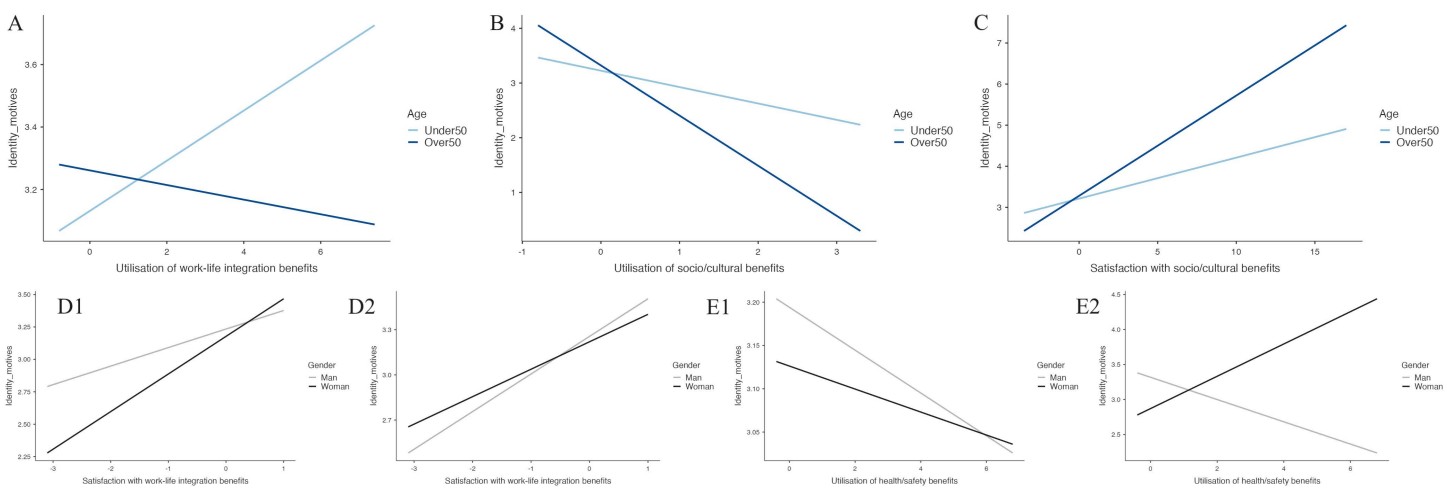

**Fig 3. The Moderating Effect in Study 2.**

Regarding the health/safety benefits, a significant effect emerged when the three-way interaction between gender, age and the quantitative indicator of health/safety benefits is considered. The simple slope analysis suggested that although from men ($b=-.02$, SE $=.05$, $t=-.37$, $p=.71$) and women ($b=-.01$, SE $=.05$, $t=-.23$, $p=.82$) under 50 (Fig 3 E 1) the association of the quantity of health/safety benefits with identity needs was not significative, when over 50 employees were considered (Fig 3 E 2), the association was slightly positive for women ($b=.23$, SE $=.13$, $t=1.85$, $p=.06$) and non-significant for men ($b=-.15$, SE $=.13$, $t=-1.34$, $p=.18$).

**O4: Testing a mediation model in which employees' identity needs mediate the effect of employee benefits on job satisfaction.** Across the four categories of benefits, the quality of the benefits used was associated with higher identity needs fulfilment, which in turn reflected higher job satisfaction.

When the work-life integration benefits were considered (Fig 4 A, RMSEA $=.06$, CFI $=.95$, TLI $=.87$), both the indirect effects of the quality of the work-life integration benefits ($b=.22$, SE $=.02$, $z=11.41$, $p<.001$) and the quantity of work-life integration benefits ($b=.04$, SE $=.02$, $z=2.34$, $p=.02$) were statistically significant. The mediation effects were partial, as the direct effects of the quality of work-life integration benefits on job satisfaction ($b=.13$, SE $=.03$, $z=3.92$, $p<001$) and the quantity of work-life integration benefits ($b=.06$, SE $=.03$, $z=2.14$, $p=.03$) remain significant.

For health/safety benefits (RMSEA $=.04$, CFI $=.98$, TLI $=.94$), Fig 4 B reports the regression coefficient of the entire model. We tested the indirect effect of the quantitative and the qualitative indicators of the health/safety benefits on job satisfaction through identity needs fulfilment. Only the indirect effect which considered the quality of the health/safety benefits as the exogenous variable was statistically significant, $b=.20$, SE $=.03$, $z=7.11$, $p<.001$ (quantity of health/safety

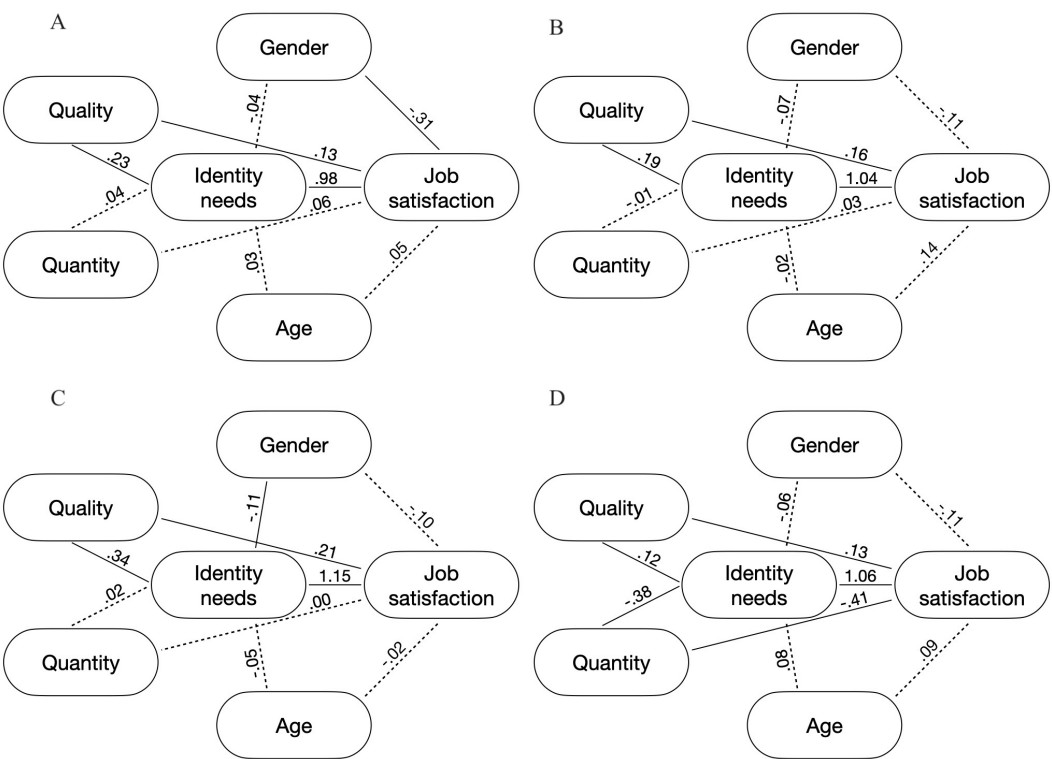

**Fig 4. Mediation Models in Study 2.** Dashed lines indicate non-significant effects.

benefits: $b = -.01$, SE = .03, $z = -.16$, $p = .87$). The mediation effect was partial, as the direct effect of the quality of the health/safety benefits remains significant, $b = .16$, SE = .04, $z = 3.81$, $p < .001$.

Concerning financial benefits (RMSEA = .06, CFI = .97, TLI = .92), Fig 4 C reports the regression coefficient of the entire model. We tested the indirect effect of the quantitative and qualitative indicators of the financial benefits on job satisfaction through identity needs fulfilment. Only the indirect effect of the quality of the financial benefit was statistically significant, $b = .39$, SE = .03, $z = 11.92$, $p < .001$ (quantity of financial benefits: $b = .02$, SE = .02, $z = 1.44$, $p = .15$). The mediation effect was partial, as the direct effects of the quality of the financial benefits remain significant, $b = .21$, SE = .05, $z = 4.70$, $p < .001$.

Interestingly, when socio/cultural benefits were considered (RMSEA = .64, CFI = .15, TLI = −1.14), quantitative and qualitative indicators showed different effects. Fig 4 D reports the regression coefficient of the entire model. We tested the indirect effect of the quantitative and qualitative indicators of the socio/cultural benefits on job satisfaction through identity needs fulfilment. Both the indirect effect of the quantity ($b = -.40$, SE = .03, $z = -11.91$, $p < .001$) and the quality ($b = .13$, SE = .01, $z = 13.97$, $p < .001$) of the socio/cultural benefits as the exogenous variable were statistically significant. The mediation effects were partial, as the direct effect of the quantity ($b = -.41$, SE = .05, $z = -8.77$, $p < .001$) and the quality ($b = .13$, SE = .01, $z = 10.76$, $p < .001$) of the socio/cultural benefits remain significant. When socio/cultural benefits are implied, different patterns emerge when the quantitative and the qualitative indicators are considered. If the higher quality of the socio/cultural benefits was positively associated with identity needs fulfilment, which in turn was positively associated with higher job satisfaction, the quantity of the socio/cultural benefits was negatively associated with identity needs fulfilment, which in turn was negatively associated with job satisfaction.

## Discussion

In Study 2, we aimed to replicate the findings from Study 1 and to examine whether the quantitative and the qualitative indicators of the different categories of services predict job satisfaction through identity needs.

Consistent with the results of Study 1, the quality of all the benefits categories appears to be more strongly associated with identity needs than quantity. Furthermore, considering age and gender, work-life integration benefits seem to be more strongly associated with identity needs more for employees under 50, especially women, compared to those over 50. Moreover, the quantity of health/safety services is associated with greater identity needs fulfilment among women over 50. For socio-cultural benefits, a complex pattern of results emerged: quantity seems to be negatively associated with identity needs for those over 50, but the quality of these benefits seems to be positively associated with identity needs in this age group.

Mediation analysis suggests that generally, the quality of the benefits is associated with higher job satisfaction through increased identity needs. Less consistent findings emerged when benefit quantity was considered. Indeed, the quantity seems to be associated with higher job satisfaction through the identity needs only for the category of work-life integration. Surprisingly, in Study 2, the quantity of the socio-cultural benefits shows a negative association with identity needs fulfilment which in turn negatively affected overall job satisfaction. This may be because individuals engaging more in socio/cultural activities may already experience low job satisfaction and seek these activities to explore opportunities beyond their current jobs [75]. Indeed, this finding may reflect compensatory behaviour: employees with lower organizational commitment or job satisfaction might seek greater engagement in socio-cultural activities as a coping mechanism for workplace dissatisfaction [76]. From a psychological compensation theory perspective, individuals may over-utilize recreational benefits when core job characteristics fail to meet their intrinsic motivation needs [77]. This behaviour might signal misaligned priorities between employee needs and organizational offerings, potentially indicating that fundamental workplace issues remain unaddressed [30]. However, these findings are limited to Study 2, thus further study should address possible contextual intervening aspects in explaining this inconsistency between Study 1 and Study 2 findings.

Overall, Study 2 confirms the results of Study 1, highlighting that organisations should prioritize benefit quality over quantity and tailor welfare services to meet employees' specific needs. Moreover, these findings provide evidence that employee benefits are important assets for job satisfaction.

## General Discussion

Employee benefits have gained increasing attention due to their positive outcomes for both employees and employers. Indeed, these programs can lead to reduced turnover rates and higher economic returns [31,32,78]. However, current research lacks a finetuned understanding of how the quantity and quality of different types of benefits impact employees across different gender and age groups. This paper addresses these gaps through two targeted studies integrating the social exchange theory with an identity-focused hypothesis.

In Studies 1 and 2, we examined the association of a quantitative and qualitative employee benefits model with individuals' identity needs and how gender and age may modulate this association using two large samples of employees. Our findings suggest that satisfaction with four categories of benefits-work-life integration, health/safety, financial and socio/cultural was better associated with participants' identity needs compared to the mere utilisation of these benefits. This result contributes to the quality versus quantity debate in HR practices [79], reinforcing that the quality of services is more useful to increase positive outcomes than quantity [80]. Therefore, it seems that rather than offering a very broad range of poorly targeted benefits, it is more important to invest in the implementation of high-quality benefits.

Furthermore, it seems that one size does not fit all. In other words, our results indicate that not all the benefits categories have the same effect for all participants. For instance, satisfaction with financial benefits is more important for men's identity compared to women's. Previous research on consumer behaviour indicates that women generally show less interest in money and possess lower materialistic values [81]. This phenomenon may be attributed to cultural influences that foster ambivalent attitudes toward money among women [82]. Therefore, this may translate into a larger distance between women's identity from finance-related opportunities compared to men. Age-related differences also emerged; employees under 50 reported higher identity needs fulfilment from the utilisation of work-life integration benefits. It is not surprising that employees under 50 derive satisfaction from their workplace when they find opportunities to manage both work and family responsibilities. Many individuals in this age group are raising young or teenage children, making perceived organizational support for work-life balance crucial [83]. This seems to be especially true for women under 50, for who work-life integration benefits was more strongly associated with their identity needs compared to other groups. The work-life integration pattern aligns with role strain theory [84]: younger women face simultaneous career establishment and family formation pressures, making organizational support for work-life balance particularly identity-affirming.

On the other hand, employees over 50 benefit more from socio/cultural and health/safety services. This relationship is particularly pronounced for women in this age group. These results are in line with [11] showing that older employees have needs beyond financial and work-life integration. At this stage, many have achieved significant life milestones, such as home ownership and raising children [85]. Consequently, financial concerns and managing childcare are less central, allowing them to derive greater satisfaction from the additional services offered by their employer, which enhance their socio/cultural and health/safety needs. Regarding the stronger association between satisfaction with socio-cultural benefits and identity needs in women over-50, this finding aligns closely with socioemotional selectivity theory [86], which posits that older adults increasingly prioritize emotionally meaningful social connections. For women, this effect may be amplified by a greater reliance on well-functioning workplace social activities as a copying strategy to combat potential isolation from age-gender intersectional stereotypes [64]. The lack of gender differences among younger employees seems to suggest that life-stage priorities, rather than inherent gender preferences, drive these differential benefit valuations.

Similarly, the health/safety utilization pattern (women over 50 showing positive association) reflects accumulated health concerns and caregiving responsibilities typical of this demographic intersection. In line with recent data on the health status of Italian older adults [87], women over 50 generally present high life expectancy but with major physical and mental

health problems compared to men. For women over 50, accessing high quality health benefits signals organizational recognition of an important need, and thereby fosters identity need fulfilment.

Importantly these results suggest that women over 50 seem to be particularly well-positioned to benefit from tailored employee benefits programs, as they are more positively associated with greater satisfaction with identity needs and, consequently, higher job satisfaction. This underscores the importance of designing welfare initiatives that address the specific needs of this demographic group.

Study 2 investigated whether identity processes mediate the relationship between employee benefits and job satisfaction. Consistent with previous literature [78], our study reaffirms the link between identity needs and job satisfaction, indicating that fulfilling identity needs fosters this outcome [88]. Notably, only satisfaction with services, not their utilisation, was associated with higher job satisfaction through identity needs, with the only exception of the utilisation of work-life integration services. These results emphasise once more the importance of quality over quantity in employee benefits programs. Organisations should focus on providing a few well-structured benefits rather than a broad array of services to maximise employee satisfaction within the company. Employers and HR practitioners should consider these findings carefully, as previous literature suggests that job satisfaction enhances firm economic performance [30–32].

Our research reveals that the relationship between organizational benefits and job satisfaction is more complex than previously theorized through social exchange perspectives alone [15]. Drawing from the motivated identity construction model [23,24], we found that identity needs serve as a significant mediating mechanism between organizational benefits and job satisfaction. This integrative lens advances existing frameworks in three ways that clarify how benefits work beyond reciprocity obligations. First, it explains why benefit quality consistently outperforms quantity in predicting outcomes: satisfying, meaningful benefits are more likely to fulfil core identity motives, whereas merely offering or using more benefits does not necessarily address those motives in employees' lived experience. Second, it accounts for demographic heterogeneity (e.g., stronger identity effects of financial benefits among men and socio/cultural benefits among women over 50) by recognizing that identity priorities vary across life stage and social roles, which standard exchange models are not designed to capture. Third, it identifies identity needs as the mechanism linking benefits to job satisfaction, as shown by partial mediation across categories, indicating that benefits do more than elicit felt obligations: they shape how employees construe and value their membership, thereby sustaining engagement. This identity-based mechanism aligns with Tyler and Blader's [89] group engagement model, which proposes that group practices signalling fair treatment enhance individuals' identification with the group. In line with their framework, organizations that demonstrate employee value through comprehensive benefits programs foster stronger organizational identification. Such practices of fair treatment signal that organizational membership is prestigious and worthy of pride, a place where they belong, and so on, encouraging employees to integrate their organizational affiliation into their self-identity. This integrated approach signals genuine organizational investment in employee well-being [90,91], fostering positive perceptions of the company and strengthening retention intentions [92]. Enhanced employee retention, in turn, contributes to improved job performance [93], creating a virtuous cycle that benefits both employees' economic and psychological well-being and organizational success.

These studies have important methodological limitations. Because mediation analyses based on cross-sectional data cannot establish causality and may yield biased indirect and direct effect estimates that diverge from longitudinal parameters and overstate mediation, our mediation results are interpreted as associational [94,95]. At the same time, when grounded in strong theory and reported transparently, cross-sectional mediation can provide informative evidence about associations [96]. Consistent with this approach, the present study, anchored in well-established identity-based perspective and social exchange theory, adopts an associational interpretation while acknowledging these methodological constraints. To better identify temporal ordering among benefits, identity needs, and job satisfaction, future research should use longitudinal designs that model temporal precedence and stability (e.g., autoregressive panel or cross-lagged frameworks) and, where feasible, field experiments with follow-up assessments of the same participants over time.

Moreover, Study 2 assessed job satisfaction with a single global item, which may reduce reliability compared with multi-item scales and limits tests of internal consistency. Nevertheless, meta-analytic evidence indicates that single-item overall job satisfaction shows substantial convergence with validated multi-item scales, supporting its use in large field studies when constraints require brevity and when interpretation is appropriately cautious [97]. We thus report this as a limitation and encourage future research to include multi-item measures to enable reliability estimation and construct-level sensitivity analyses.

We did not model interactions between benefit categories, which is a scope limitation in light of research showing that human resource practices can operate as complementary 'bundles' with synergistic or substitutive effects [98]. For instance, work-life integration and health/safety initiatives may jointly satisfy identity needs more than either alone (complementarity), whereas extensive socio/cultural offerings might substitute for weaker financial supports in some worker segments (substitution), potentially explaining heterogeneity in observed associations. Future studies should test interaction hypotheses and evaluate cross-category interactions (e.g., multiplicative terms or multigroup structural models), balancing theoretical specificity with controls for multiple testing and model parsimony.

Finally, while our study controlled for demographic variables, other contextual variables could play a role in shaping the effects of the impact of organizational benefits and give insight on contradictory findings between Study 1 and Study 2 (i.e., on the relationship between the quantity of socio/cultural benefits and satisfaction with identity needs). For example, research on organizational culture has shown that the utilisation of certain benefits, particularly those related to work-family flexibility, may be perceived by supervisors as reduced organizational commitment, potentially deterring employees from accessing these resources [99,100]. Thus, future research should consider that these contextual aspects may interfere with the positive effect of employee benefits on work outcomes.

These limitations reflect several constraints that research within organizations generally implicates. The cross-sectional design was required due to partner companies' operational timelines, though we employed multilevel modelling to account for firm-level clustering and reduce standard error bias [101]. Age was dichotomized at 50 years per HR department requirements, aligning with established career-stage frameworks while facilitating practical interpretation [11]. Job satisfaction was measured with a validated single-item scale ($r = .63$ convergent validity with multi-item measures) to minimize response burden in large-scale organizational surveys [97]. These decisions balanced methodological rigor with field research realities. Despite these limitations, the large sample size, multi-firm design, and real-world context provide valuable ecological validity that complements laboratory-based findings in this domain.

## Conclusions

Organisations are increasingly investing in employee benefits as a strategy to enhance overall job satisfaction and well-being, which, in turn, leads to lower turnover rates and improved economic performance. However, limited research has focused on the identity mechanisms underlying the relationship between employee benefits and job satisfaction while also accounting for the advantages of tailoring welfare programs for specific demographic groups. Our research provides valuable insights into these dynamics, emphasizing the critical role of identity needs in linking welfare programs to job satisfaction. By providing preliminary evidence of the importance of quality over quantity in welfare offerings and the varying impact of different services across gender and age groups, our findings offer a nuanced understanding that can help organisations better tailor their welfare strategies. Specifically, women over 50 benefit the most from welfare programs that are tailored to their unique needs.

In an era of increasing workforce diversity and intense competition for talent, understanding the identity-based mechanisms of employee well-being becomes strategically crucial. When organizations design benefits that address employees' identity needs, they create sustainable advantages that benefit both the workforce and the organization. This dual benefit creates a positive feedback loop, enhancing both employee well-being and organizational sustainability.

## Supporting information

**S1 Text. Preliminary findings and ANOVA omnibus effect tables.**
(DOCX)

## Author contributions

**Conceptualization:** Alessia Valmori, Claudia Manzi.

**Data curation:** Alessia Valmori.

**Formal analysis:** Alessia Valmori.

**Funding acquisition:** Claudia Manzi.

**Methodology:** Alessia Valmori, Eleonora Reverberi, Claudia Manzi.

**Supervision:** Claudia Manzi.

**Visualization:** Alessia Valmori.

**Writing – original draft:** Alessia Valmori.

**Writing – review & editing:** Eleonora Reverberi, Claudia Manzi.

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
