## [Decision Letter · Decision Letter 0]

28 Aug 2025

Dear Dr. Valmori,

Thank you for submitting your manuscript to PLOS ONE. After careful consideration, we feel that it has merit but does not fully meet PLOS ONE’s publication criteria as it currently stands. Therefore, we invite you to submit a revised version of the manuscript that addresses the points raised during the review process.

We look forward to receiving your revised manuscript.

Kind regards,

Ali Junaid Khan, PhD

Academic Editor

PLOS ONE

Reviewers' comments:

Reviewer's Responses to Questions

**Comments to the Author**

1. Is the manuscript technically sound, and do the data support the conclusions?

Reviewer #1: Yes

Reviewer #2: Partly

2. Has the statistical analysis been performed appropriately and rigorously?

Reviewer #1: Yes

Reviewer #2: Yes

3. Have the authors made all data underlying the findings in their manuscript fully available?

Reviewer #1: Yes

Reviewer #2: Yes

4. Is the manuscript presented in an intelligible fashion and written in standard English?

Reviewer #1: Yes

Reviewer #2: Yes

Reviewer #1: The manuscript presents a valuable and well-executed empirical study on the role of employee benefits in fulfilling identity needs and enhancing job satisfaction, with important implications for HR policy. The integration of social exchange theory and identity-based mechanisms is theoretically strong, and the large datasets provide robust support for the conclusions. I recommend minor revisions to literature (Is it reasonable to repeat certain terms multiple times and continually return to them throughout the text - like job satisfaction?), write whole word for abbreviations (ESG). Expand the discussion of methodological limitations, particularly the cross-sectional design and the use of a single-item job satisfaction measure, and consider discussing potential interactions between benefit categories. Overall, this is a solid contribution to the literature.

Reviewer #2: Thank you for the opportunity to review your manuscript. The study is ambitious, with a large dataset and a meaningful research question. At the same time, I believe the paper needs significant revisions before it is suitable for publication. My detailed comments are as follows:

1) Theoretical Contribution

- The paper’s novelty needs to be articulated more clearly. The claim that benefit quality matters more than benefit quantity has been supported in prior literature; thus, you should clarify how your study advances theory beyond this general claim.

- The integration of social exchange theory with identity-based mechanisms is promising, but the discussion does not fully show how this combination provides new insights compared with existing frameworks.

2) Methodology

- Both studies are cross-sectional surveys. While the large sample sizes are a strength, causal inference remains weak. Consider adding robustness checks, alternative model specifications, or at least a stronger justification for the analytic choices.

- The mediation analysis in Study 2 should be interpreted with caution. Cross-sectional data cannot definitively establish mediation. Please temper your claims and clearly state the limits of your design.

3) Results and Interpretation

- Some moderation and interaction effects (particularly three-way interactions) are complex but only superficially interpreted. More in-depth theoretical explanation is needed rather than descriptive reporting.

- The negative association between the quantity of socio-cultural benefits and identity/job satisfaction is particularly interesting. However, the explanation provided is underdeveloped. This deserves a fuller discussion, including alternative interpretations.

4) Writing and Structure

- The paper is overly lengthy. The literature review in particular could be condensed, focusing only on the most relevant theories and prior findings.

- Results are presented in detail, but the discussion does not sufficiently highlight the key takeaways. Consider moving some secondary findings to an appendix.

- Figures and tables are useful, but the narrative around them should be streamlined to emphasize the most important contributions.

This manuscript has promise, particularly due to the impressive dataset and the attention to demographic moderators. However, substantial revision is necessary to clarify the theoretical contribution, address methodological concerns, provide deeper interpretations of complex findings, and improve the clarity and conciseness of the writing. I encourage you to work out these points and resubmit.

**Do you want your identity to be public for this peer review?** For information about this choice, including consent withdrawal, please see our Privacy Policy

Reviewer #1: **Yes: ** Nejc Bernik

Reviewer #2: No

---

## [Author Response · Author response to Decision Letter 1]

25 Sep 2025

Reviewer #1: The manuscript presents a valuable and well-executed empirical study on the role of employee benefits in fulfilling identity needs and enhancing job satisfaction, with important implications for HR policy. The integration of social exchange theory and identity-based mechanisms is theoretically strong, and the large datasets provide robust support for the conclusions. I recommend minor revisions to literature (Is it reasonable to repeat certain terms multiple times and continually return to them throughout the text - like job satisfaction?), write whole word for abbreviations (ESG). Expand the discussion of methodological limitations, particularly the cross-sectional design and the use of a single-item job satisfaction measure and consider discussing potential interactions between benefit categories.

Overall, this is a solid contribution to the literature.

AU: We thank the reviewer for the constructive recommendations regarding literature presentation and terminology usage. To reduce unnecessary redundancy, we streamlined the literature narrative to avoid repeatedly returning to the construct of job satisfaction across sections, consolidating overlapping sentences and replacing proximate repetitions with precise references to the construct only where needed for clarity and consistency with our theoretical model.

We have also expanded the limitations to explicitly acknowledge the constraints of a cross‑sectional design for causal and mediation claims, noting well‑documented risks of bias in cross‑sectional mediation estimates and clarifying that all indirect effects are interpreted as associational pending longitudinal or experimental tests. We also added a measurement limitation for the single‑item job satisfaction indicator, recognizing reduced reliability compared to multi‑item scales while citing meta‑analytic evidence that single‑item overall job satisfaction shows substantial convergence with validated multi‑item measures and can be acceptable in large‑scale field studies when used transparently and with caution. Finally, we introduce a paragraph motivating potential interactions between benefit categories (e.g., complementary or substitutive “bundle” effects) and outline this as a theoretically grounded avenue for future analysis rather than a post‑hoc exploratory exercise in the present paper, given model complexity and multiple‑testing risks. The revised limitations section is on pages 29-31.

Reviewer #2: Thank you for the opportunity to review your manuscript. The study is ambitious, with a large dataset and a meaningful research question. At the same time, I believe the paper needs significant revisions before it is suitable for publication. My detailed comments are as follows:

1 - The paper’s novelty needs to be articulated more clearly. The claim that benefit quality matters more than benefit quantity has been supported in prior literature; thus, you should clarify how your study advances theory beyond this general claim.

AU: Thank you for this helpful comment. The Introduction now explicitly frames our contribution as a refinement of the “quality > quantity” claim: we propose to test whether the advantage of benefit quality over mere utilisation generalises across welfare categories and across age/gender in large, multi‑firm datasets, rather than only in aggregate. We also briefly justify why this matters by identifying scope conditions and guiding resource allocation toward fewer, higher‑quality benefits that more reliably enhance identity needs and job satisfaction:

“..we aim to extend the ‘quality > quantity’ claim by testing whether the advantage of benefit quality over mere utilisation holds across different welfare categories (work–life integration, health/safety, financial, socio/cultural) and across key sociodemographic profiles (age and gender). Breaking down the analyses by type of benefit and socio-demographic classes is an important innovative element that provides valuable insights to help organisations target scarce resources toward fewer, higher‑quality benefits that reliably improve workers’ outcomes for specific groups, avoiding costly, low‑impact expansions in benefit quantity that may not translate into meaningful employee outcomes.”

2 - The integration of social exchange theory with identity-based mechanisms is promising, but the discussion does not fully show how this combination provides new insights compared with existing frameworks.

AU: We thank the reviewer for this valuable feedback, which prompted us to clarify the unique contributions of our integrative approach. We have expanded the General Discussion section to explicitly articulate how combining social exchange theory with identity-based mechanisms advances beyond existing frameworks. Specifically, we now explain that this integrative lens advances existing frameworks in three distinctive ways that clarify how benefits work beyond reciprocity obligations:

“This integrative lens advances existing frameworks in three ways that clarify how benefits work beyond reciprocity obligations. First, it explains why benefit quality consistently outperforms quantity in predicting outcomes: satisfying, meaningful benefits are more likely to fulfil core identity motives, whereas merely offering or using more benefits does not necessarily address those motives in employees’ lived experience. Second, it accounts for demographic heterogeneity (e.g., stronger identity effects of financial benefits among men and socio/cultural benefits among women over 50) by recognizing that identity priorities vary across life stage and social roles, which standard exchange models are not designed to capture. Third, it identifies identity needs as the mechanism linking benefits to job satisfaction, as shown by partial mediation across categories, indicating that benefits do more than elicit felt obligations: they shape how employees construe and value their membership, thereby sustaining engagement.”

3 - Both studies are cross-sectional surveys. While the large sample sizes are a strength, causal inference remains weak. Consider adding robustness checks, alternative model specifications, or at least a stronger justification for the analytic choices.

AU: We have addressed the concern about analytical choices by adding explicit justification in the General Discussion section. The paragraph acknowledges that our design decisions (cross-sectional format, dichotomous age, single-item job satisfaction) were necessitated by organizational constraints while maintaining methodological rigor through multilevel modeling and validated measures. We emphasize that these represent common trade-offs in field research where ecological validity complements controlled studies:

“These limitations reflect several constraints that research within organizations generally implicates. The cross-sectional design was required due to partner companies’ operational timelines, though we employed multilevel modelling to account for firm-level clustering and reduce standard error bias [101]. Age was dichotomized at 50 years per HR department requirements, aligning with established career-stage frameworks while facilitating practical interpretation [11]. Job satisfaction was measured with a validated single-item scale (r = .63 convergent validity with multi-item measures) to minimize response burden in large-scale organizational surveys [97]. These decisions balanced methodological rigor with field research realities. Despite these limitations, the large sample size, multi-firm design, and real-world context provide valuable ecological validity that complements laboratory-based findings in this domain.”

4 - The mediation analysis in Study 2 should be interpreted with caution. Cross-sectional data cannot definitively establish mediation. Please temper your claims and clearly state the limits of your design.

AU: Thank you for your comment which align also with R1’s concern. In this regard we have used more adequate terms for expressing association instead of predictive effects on both the presentation of the Results and in the Discussion sessions. We also added a limitation paragraph in the General Discussion in which we more explicitly state this point:

“Because mediation analyses based on cross‑sectional data cannot establish causality and may yield biased indirect and direct effect estimates that diverge from longitudinal parameters and overstate mediation, our mediation results are interpreted as associational [94,95]. At the same time, when grounded in strong theory and reported transparently, cross‑sectional mediation can provide informative evidence about associations [96]. Consistent with this approach, the present study, anchored in well‑established identity-based perspective and social exchange theory, adopts an associational interpretation while acknowledging these methodological constraints. To better identify temporal ordering among benefits, identity needs, and job satisfaction, future research should use longitudinal designs that model temporal precedence and stability (e.g., autoregressive panel or cross‑lagged frameworks) and, where feasible, field experiments with follow‑up assessments of the same participants over time.”

5 - Some moderation and interaction effects (particularly three-way interactions) are complex but only superficially interpreted. More in-depth theoretical explanation is needed rather than descriptive reporting.

AU: We have substantially enhanced our interpretation of the three-way interactions by integrating established psychological theories. We now apply socioemotional selectivity theory to explain why women over 50 derive greater identity fulfillment from socio-cultural benefits, and role strain theory to clarify work-life integration and health/safety patterns across demographic groups. In the General Discussion session, we added the following paragraph:

“Regarding the stronger association between satisfaction with socio-cultural benefits and identity needs in women over-50, this finding aligns closely with socioemotional selectivity theory [86], which posits that older adults increasingly prioritize emotionally meaningful social connections. For women, this effect may be amplified by a greater reliance on well-functioning workplace social activities as a copying strategy to combat potential isolation from age-gender intersectional stereotypes [64]. The lack of gender differences among younger employees seems to suggest that life-stage priorities, rather than inherent gender preferences, drive these differential benefit valuations. Similarly, the health/safety utilization pattern (women over 50 showing positive association) reflects accumulated health concerns and caregiving responsibilities typical of this demographic intersection. In line with recent data on the health status of Italian older adults [87], women over 50 generally present high life expectancy but with major physical and mental health problems compared to men. For women over 50, accessing health benefits signals organizational recognition of their dual burden as both older employees and family caregivers, thereby fulfilling identity need.”

6 - The negative association between the quantity of socio-cultural benefits and identity/job satisfaction is particularly interesting. However, the explanation provided is underdeveloped. This deserves a fuller discussion, including alternative interpretations.

AU: We thank the reviewer for this valuable observation. We have addressed this concern by incorporating a theoretical framework to explain this counterintuitive finding. Specifically, we added a discussion of compensatory behavior theory, which suggests that employees with lower job satisfaction may seek greater engagement in socio-cultural activities as a coping mechanism for workplace dissatisfaction. This perspective, supported by psychological compensation theory (Deci & Ryan, 2000), provides a plausible explanation for why employees might over-utilize recreational benefits when core job characteristics fail to meet their intrinsic needs:

“Indeed, this finding may reflect compensatory behaviour: employees with lower organizational commitment or job satisfaction might seek greater engagement in socio-cultural activities as a coping mechanism for workplace dissatisfaction [76]. From a psychological compensation theory perspective, individuals may over-utilize recreational benefits when core job characteristics fail to meet their intrinsic motivation needs [77]. This behaviour might signal misaligned priorities between employee needs and organizational offerings, potentially indicating that fundamental workplace issues remain unaddressed [30].”

7 - The paper is overly lengthy. The literature review in particular could be condensed, focusing only on the most relevant theories and prior findings.

AU: We agree with the reviewer that the manuscript was overly lengthy. We have condensed sections in the literature review to avoid excessive argumentation on specific points. We removed the historical excursus on welfare, streamlined the presentation of benefit categories, and condensed the analysis of gender and age effects to focus on key findings.

8 - Results are presented in detail, but the discussion does not sufficiently highlight the key takeaways. Consider moving some secondary findings to an appendix.

AU: We have restructured the General Discussion section by presenting the main findings in a clearer and more concise order. We begin with the discussion of quality versus quantity of benefits, followed by an analysis of the moderating effects. Subsequently, we address the association between benefits and job satisfaction. Finally, we discuss the integration of social exchange theory with the identity-based perspective. We have also moved the Tables presenting the ANOVA Omnibus Effects to the Supporting Information, while maintaining the presentation of the regression coefficients in the Result session.

9 - Figures and tables are useful, but the narrative around them should be streamlined to emphasize the most important contributions.

AU: We thank the reviewer for this suggestion. We have streamlined and simplified all the captions of tables and figures, ensuring they are more concise and directly highlight the paper’s key contributions.

---

## [Decision Letter · Decision Letter 1]

30 Oct 2025

One Size Does Not Fit All:

Nurturing Identity Needs and Job Satisfaction Through Employee Benefits Across Gender and Age

PONE-D-25-38613R1

Dear Dr. Valmori,

We’re pleased to inform you that your manuscript has been judged scientifically suitable for publication and will be formally accepted for publication once it meets all outstanding technical requirements.

Kind regards,

Ali Junaid Khan, PhD

Academic Editor

PLOS ONE

Additional Editor Comments (optional):

Reviewers' comments:

Reviewer's Responses to Questions

**Comments to the Author**

Reviewer #1: All comments have been addressed

Reviewer #3: All comments have been addressed

2. Is the manuscript technically sound, and do the data support the conclusions?

Reviewer #1: Yes

Reviewer #3: No

3. Has the statistical analysis been performed appropriately and rigorously?

Reviewer #1: Yes

Reviewer #3: Yes

4. Have the authors made all data underlying the findings in their manuscript fully available?

Reviewer #1: Yes

Reviewer #3: Yes

5. Is the manuscript presented in an intelligible fashion and written in standard English?

Reviewer #1: Yes

Reviewer #3: Yes

Reviewer #1: The authors have thoroughly revised the manuscript in accordance with the my and other reviewer comments. All major concerns have been satisfactorily addressed, and the quality of the manuscript has improved. The data described in the main text are appropriate and clearly presented. It is a positive improvement that the authors moved the lengthy tables to the appendix, which improves the readability and flow of the main manuscript. Additionally, the conclusion is now well supported by the presented results and aligns with the main objectives of the study. I recommend the revised version for publication.

Reviewer #3: The manuscript presents a good and valuable, well defined contribution on role of employee benefits. However, I do recommed minor revisions.

The further elaboration of cross sectional and single item measures is reasonable. Possible self-report and common method biases would be briefly mentioned because it would be helpful to increase methodological transparency.

The presentation is clear, but captions can be reduced even more to highlight major insights as opposed to the process.

**Do you want your identity to be public for this peer review?** For information about this choice, including consent withdrawal, please see our Privacy Policy

Reviewer #1: No

Reviewer #3: **Yes: ** Shahar Yar, DBA

---

## [Editor Report · Acceptance letter]

PONE-D-25-38613R1

PLOS ONE

Dear Dr. Valmori,

I'm pleased to inform you that your manuscript has been deemed suitable for publication in PLOS ONE. Congratulations! Your manuscript is now being handed over to our production team.

Kind regards,

on behalf of

Dr Ali Junaid Khan

Academic Editor

PLOS ONE